# Magnitude of optimal access to ANC and its predictors in Ethiopia: Multilevel mixed effect analysis of nationally representative cross-sectional survey

Wubshet Debebe Negash[1]*, Tadele Biresaw Belachew[1], Samrawit Mihret Fetene[1], Banchilay Addis[1], Tsegaw Amare[1], Atitegeb Abera Kidie[2], Abel Endawkie[3], Alebachew Ferede Zegeye[4], Tadesse Tarik Tamir[5], Sisay Maru Wubante[6], Elsa Awoke Fentie[7], Desale Bihonegn Asmamaw[7]

1 Department of Health Systems and Policy, Institute of Public Health, College of Medicine and Health Sciences, University of Gondar, Gondar, Ethiopia, 2 School of Public Health, College of Health Science, Woldia University, Woldia, Ethiopia, 3 Department of Epidemiology and Biostatistics, School of Public Health, College of Medicine and Health Science, Wollo University, Dessie, Ethiopia, 4 Department of Medical Nursing, School of Nursing, College of Medicine and Health Sciences, University of Gondar, Gondar, Ethiopia, 5 Department of Pediatric and Child Health Nursing, School of Nursing, College of Medicine and Health Sciences, University of Gondar, Gondar, Ethiopia, 6 Department of Health Informatics, Institute of Public Health, College of Medicine and Health Sciences, University of Gondar, Gondar, Ethiopia, 7 Department of Reproductive Health, Institute of Public Health, College of Medicine and Health Sciences, University of Gondar, Gondar, Ethiopia

* wubshetdn@gmail.com

**Data Availability Statement:** The study used publicly available data, which can be found at the following link: https://dhsprogram.com/data/

## Abstract

### Background

Optimal access to ANC, such as the first ANC visit at first trimester, four or more ANC visits, and skilled health care provider can significantly reduce maternal mortality in an inclusive way. Previous studies conducted in Ethiopia on optimal ANC are restricted to frequencies of ANC visit. Therefore, the aim of this study was to assess the magnitude of optimal ANC access as a comprehensive way and its predictors among pregnant women in Ethiopia.

### Methods

Secondary data source from a recent demographic and health survey was used for analysis. This study includes a weighted sample of 4771 pregnant women. A multilevel mixed-effect binary logistic regression analyses was done to identify both the individual and community level factors. Odds ratio along with the 95% confidence interval was generated to identify the predictors of optimal access to ANC. A p-value less than 0.05 was declared as statistical significant.

### Results

In Ethiopia, one in five (20%) pregnant women had optimal access to antenatal care. Regarding the factors at individual level, pregnant women aged 25–34 years [aOR = 1.58, 95% CI = 1.23–2.03] and 35–49 years [aOR = 2.04, 95% CI = 1.43–2.89], those who had

dataset_admin/login_main.cfm?CFID=
39421058&CFTOKEN=a8b8a36f1fb27230-
E89DAEA4-D47B-719A 9AD1F13D7D93EF8A.

**Funding:** The authors received no specific funding
for this work.

**Competing interests:** The authors declare that they
have no competing interests.

**Abbreviations:** ANC, Antenatal care; AOR, Adjusted
Odds Ratio; CSA, Central Statistical Agency; DHS,
Demographic Health Survey; EAs, Enumeration
Areas; EDHS, Ethiopian Demographic and Health
Survey; ICC, Intra-class Correlation Coefficient;
MOR, Median Odds Ratio; PCV, Proportional
Change in Variance; SD, Standard Deviation; WHO,
World Health Organization.

educated primary [aOR = 1.67, 95% CI = 1.33–2.09], secondary and higher [aOR = 1.81, 95% CI = 1.15–2.85], Primipara [aOR = 2.45, 95% CI = 1.68–3.59] and multipara [aOR = 1.48, 95% CI = 1.11–1.98] had higher odds of accessing optimal ANC. With the community level factors, the odds of optimal access to ANC was higher among pregnant women who lived in urban area [aOR = 2.08, 95% CI = 1.33–3.27], whereas, lower odds of optimal ANC access among those pregnant women who reported distance to the health facility as a big problem [aOR = 0.78, 95% CI = 0.63–0.96].

## Conclusion and recommendation

The study concludes that in Ethiopia, optimal access to ANC was low. The study identified that both individual and community level factors were predictors for optimal ANC access. Therefore, the Ethiopian government should intensify extensive education on ANC in a comprehensive way. Moreover, especial attention from the Ethiopian ministry of health for those women who reported distance as a big problem and for rural resident women is mandatory.

## Background

Antenatal care (ANC) is a care structured for the mother, infant, and child based on the following recommended basic packages: The detection and treatment of disorders (such as anemia, abnormal lying, diabetes, syphilis, and hypertension); the provision of preventive interventions (like tetanus vaccinations and insecticide-treated bed nets); as well as the advice on diet, hygiene, HIV status, birth, emergency preparedness, and baby care and feeding [1, 2].

In addition to the single measures of antenatal care visit, it can be assessed as a whole by combining initiation of first ANC, frequencies of ANC visit and type of ANC provider in order to get a more comprehensive picture of the benefits of antenatal care [3]. Evidences indicated that early initiation of first ANC enables to increase the probability of number of ANC contacts [4–6]. Correspondingly, it increases the likelihood of skilled birth attendance [7] and provides health workers with a chance to educate pregnant women on essential pregnancy care to reduce malnutrition, stillbirths, maternal mortality, and neonatal mortality [8].

It is estimated that more than half of maternal deaths worldwide are caused by pregnancy-related complications, and that over 90% of these deaths occur in low and middle income countries [9]. There have been an estimated 295,000 pregnancies and childbirth related deaths since 2017, with 94% of those deaths occurring in low and lower middle-income countries [10]. An estimated 14,000 maternal deaths occurred in Ethiopia in 2017, contributing to a maternal mortality rate of 402 deaths per 100,000 live births [10]. A large number of the above maternal deaths could be prevented by initiation of first ANC at first trimester, skilled birth attendance and increased antenatal care [9].

There is considerable evidence that the effectiveness of ANC is strongly influenced by the essential services covered during visits [11, 12]. Because in the unstable and unpredictable labor and delivery period, providing universal care is more challenging than in the longer, more stable antenatal period [3, 13]. Increased ANC coverage and quality of healthcare can avert 71% of neonatal mortality, 33% of stillbirths, and 54% of maternal mortality in low and middle income countries (LMICs) [14]. There is evidence that a proper access to ANC can reduce maternal mortality by up to 8 per 1000 live births [9].

Interventions to reduce maternal mortality such as Universal access to reproductive health services (ANC and others) [15], family planning 2020 [16, 17], the Ethiopian government considerations of maternal health care services through giving priority as a political agenda to reduce the maternal mortality below 267 deaths per 100,000 live births [18] had been executed. However, the maternal [19–21], and neonatal mortality rates continue to rise in developing nations. Undoubtedly needs an improved optimal access to ANC services with an attention for the timing of first ANC, frequencies of ANC and skilled ANC attendant [11, 12].

In most previous studies conducted in Ethiopia [22, 23], only the number of ANC visits were considered optimal ANC. However, in addition to the single (frequencies of ANC visit) measures of antenatal care, it can be assessed as a whole in order to get a more comprehensive picture of the benefits of antenatal care [3]. It is important to evaluate access to ANC at a national level with the recent standardized Ethiopian Demographic and Health Survey (EDHS) data. Hence, this study examines the magnitude of optimal access to ANC by focusing on three key indicators: timing of first ANC initiation, frequency of ANC, and type of ANC provider, individual and community level predictors with a multilevel mixed effect approach.

## Methods

### Study settings and data source

A cross-sectional study of Ethiopian Demographic and Health survey (EDHS) data was used for this study. Ethiopia is located in the horn of Africa and administratively divided in to nine ethnic based regional states (Tigray, Afar, Amhara, Oromia, Benishangul, Gambela, South Nation Nationalities and People Region (SNNPR), Harari, and Somali), and two administrative cities (Addis Ababa and Dire-Dawa), 611 Districts, and 15,000 Kebeles. In partnership with the Ethiopian Public Health Institute (EPHI) and the Federal Ministry of Health (FMoH), the Central Statistical Agency (CSA) conducted the survey from January 18 to June 27, 2016. This study used the women's recode (IR file) data set and extracted the outcome and predictor variables. Here is a link to the free data set that can be downloaded: https://dhsprogram.com/data/available-datasets.cfm. The EDHS employs a two-stage stratified sampling technique [24]. In this study, a total weighted sample of 4771 pregnant women aged 15–49 years were included. Pregnant women who had not received ANC visits were excluded.

### Outcome variable creation

Based on the factors and outcomes associated with better ANC used in different studies and the WHO recommendations [2, 3, 8, 25–27], access to ANC was developed to determine the variable optimal access to ANC. Optimal access to ANC was constructed by combining three indicators: timing of first ANC visit, number of ANC visits, and type of provider of ANC. In EDHS the initiation of first ANC is available ranging from 0 to 10 months, then the first three months were considered as early ANC initiation coded as 1 and 0 for after 3 months. Similarly, total number of ANC contacts was presented from 1 to 20 visits. Total number of ANC was categorized into four or more visits as coded as "1" and less than four visits coded as "0" [22, 23] this is because in Ethiopia the focus ANC is still recommended. Skilled health care provider refers to the provider of ANC who provided her with ANC during her last pregnancy. In this study, experts were classified into skilled providers: doctors, clinical officers, nurses, health officers, extension workers, and midwives, coded as 1 and non-skilled providers such as traditional birth attendant, community health workers coded as 0. When multiple providers provided care to a woman, the highest skilled provider was recorded [28]. Finally, a woman was considered as optimal access to ANC if she had get all the three ANC components that is

initiated her ANC in the first trimester plus a minimum of four ANC contacts plus provided by skilled health care provider [29].

**Independent variables.** Both individual and community level variables were included in the independent variables. The individual level variables were age, occupational status of the women, educational level of the women and her husband, religion, wealth index, media exposure, number of children and parity whereas residence, distance to the health facility, region, community level media exposure, community level poverty and community level education were included from the community level factors.

Accordingly, age was grouped as 15–24, 25–34, and 35–49 years. Occupational status was recoded as working and not working. No formal education, primary education, secondary and higher education were the categories for highest educational level to the mother and her husband. In EDHS wealth index was developed by principal component analysis using durable asset ownership, housing characteristics and access to utilities. Finally the wealth index was recoded as poor, middle, and rich. Those women who were either reading newspapers/magazine, or listening radio and watching television less than once a week/at least once a week were considered as having media exposure whereas, those women who had not either reading magazine/newspaper or listening radio/ television at all was considered as having not media exposure. Variables such as place of residence (urban, rural), distance to the health facility (big problem, not big problem) were analyzed based on their categorization in the EDHS [30–33]. The community level poverty, community level education and community level media exposure were generated by aggregating the individual level factors independently at cluster level and finally, were categorized as high if the proportion is ≥50% and low if the proportion is <50% based on the national median value since these were not normally distributed [34]. All the aforementioned independent variables were included based on their practical significance for optimal ANC access.

## Statistical analyses

A multilevel logistic regression model was used to identify the association between the individual and community level factors with optimal access to ANC. Stata version 14 command "melogit" was used in fitting the models. The data was weighted (v005/1,000,000) throughout the analysis to ensure the EDHS sample representative and to obtain reliable estimates and standard errors before data analysis (Table 1). Overall, a total weighted sample of 4771 reproductive aged pregnant women were included in this study. First the distribution of optimal ANC access across the individual and community level characteristics was done. Second a graphical representation of optimal ANC access among reproductive age women was presented. Statistical significance of association at p vale < 0.05 using Pearson's chi square test of independence ($X^2$) between each of the independent variable and optimal ANC access was determined.

All the variables having a p-value less than 0.05 in bivariable analysis were used for multivariable analysis. For the multivariable analysis, adjusted odds ratios with 95% confidence intervals and a p-value of less than 0.05 were used to identify statistically significant factors associated with optimal access to ANC.

In the final step of the analysis, a multilevel mixed effect logistic regression analysis contained fixed effect variables that are all of the independent (both individual and community level variables) and random variable (cluster number) was conducted. Based on the random variable the measure of variation such as intra-class correlation coefficient (ICC) and median odds ratio (MOR) were calculated [35]. The results of the fixed effects of the model were presented as adjusted odds ratio (AOR). Accordingly, four models were fitted; null model (model

**Table 1. Distribution of optimal access to ANC across individual and community level factors of pregnant women in Ethiopia, 2016 (n = 4771).**

| Variables | Weighted(N) | Weighted (%) | Optimal access to ANC | X²(P-value) |
|---|---|---|---|---|
| **Individual level variables** | | | | |
| **Age** | | | | 5.67(0.05) |
| 15–24 | 1236 | 25.91 | 22.73 | |
| 25–34 | 2496 | 52.31 | 25.09 | |
| 35 and above | 1039 | 21.78 | 22.64 | |
| **Education of the mother** | | | | 219.1(<0.001) |
| No formal education | 2580 | 54.07 | 19.74 | |
| Primary | 1917 | 40.17 | 26.60 | |
| Secondary and higher | 275 | 5.76 | 44.86 | |
| **Current marital status** | | | | 15.99(<0.001) |
| Married | 4430 | 92.84 | 23.15 | |
| Unmarried | 342 | 7.16 | 34.31 | |
| **Occupation of the women** | | | | 1.42(0.023) |
| Working | 2351 | 49.26 | 26.24 | |
| Not working | 2421 | 50.74 | 21.72 | |
| **Religion** | | | | 67.58(<0.001) |
| Orthodox | 2080 | 42.54 | 30.84 | |
| Muslim | 1573 | 32.96 | 21.75 | |
| Protestant | 1049 | 21.99 | 14.61 | |
| Catholic and Traditional | 119 | 2.50 | 17.62 | |
| **Wealth index** | | | | 200(<0.001) |
| Poor | 1730 | 36.25 | 18.94 | |
| Middle | 996 | 20.88 | 21.40 | |
| Rich | 2045 | 42.87 | 29.42 | |
| **Parity** | | | | 103.43(<0.001) |
| 1 | 1123 | 23.53 | 30.07 | |
| 2–5 | 2576 | 53.99 | 24.32 | |
| 5 and above | 1073 | 22.49 | 16.63 | |
| **Decision maker for health service utilization** | | | | 17.33(<0.001) |
| Women | 3762 | 83.95 | 23.80 | |
| Other* | 719 | 16.05 | 20.63 | |
| **Husband education** | | | | 172.22(<0.001) |
| No formal education | 1816 | 40.53 | 20.49 | |
| Primary | 2246 | 50.12 | 22.65 | |
| Secondary and higher | 419 | 9.35 | 38.86 | |
| **Media exposure** | | | | 164.72(<0.001) |
| Yes | 2028 | 42.50 | 29.56 | |
| No | 2743 | 57.50 | 19.80 | |
| **Community level variables** | | | | |
| **Distance to the health facility** | | | | 35.64(<0.001) |
| Not big problem | 2401 | 50.33 | 21.00 | |
| Big problem | 2370 | 49.67 | 26.93 | |
| **Residence** | | | | 322.22(<0.001) |
| Urban | 875 | 18.34 | 40.26 | |
| Rural | 3896 | 81.66 | 20.28 | |
| **Community level media exposure** | | | | 125.38(<0.001) |
| High | 2264 | 47.45 | 27.17 | |

*(Continued)*

**Table 1.** (Continued)

| Variables | Weighted(N) | Weighted (%) | Optimal access to ANC | X²(P-value) |
|---|---|---|---|---|
| Low | 2507 | 52.55 | 21.04 | |
| **Community level poverty** | | | | 174.55(<0.001) |
| High | 2448 | 51.30 | 28.08 | |
| Low | 2323 | 48.70 | 19.59 | |
| **Community level education** | | | | 137.5(<0.001) |
| High | 1781 | 37.34 | 28.24 | |
| Low | 2990 | 62.66 | 21.39 | |

*Husband, relatives

0) which shows the variations in the optimal access to ANC in the absence of any independent variables. Model II contained the individual-level variables, Model III contained the community level variables, and model IV contained both the individual and community level variables [35, 36]. The equation used for fitting the multilevel logistic regression model was as follows:

$$\text{Where}: \ \text{Log}\left[\pi_{ij}/(1 - \pi_{ij})\right] = \beta_0 + \beta_1 x_{ij} + \beta_2 x_{ij} \ldots + \mu_{0j} + e_{0ij}$$

$\pi_{ij}$: The probability of optimal ANC access

$1 - \pi_{ij}$: The probability of no optimal ANC access

$\beta_1 x_{ij}$: individual and community level variables for the $i^{th}$ individual in group j, respectively. The β's are fixed coefficients indicating a unit increase in X can cause a β unit increase in probability optimal ANC access. While the $\beta_0$ is intercept that is the effect on optimal ANC access when the effect of all explanatory variables are absent. The uj shows the random effect (effect of the community on the women's optimal ANC access) for the $j^{th}$ community [37, 38].

Correspondingly, model goodness of fit was done using the deviance (-2 log likelihood). Variance inflation factor (VIF) was used to check for multicollinearity among independent variables in which the result showed no evidence of multicollinearity (mean value for the final model = 1.59) (Table 2).

## Results

### Distribution of optimal access to ANC across the individual and community level factors

Table 1 shows results on the distribution of optimal access to ANC across the individual and community level factors among pregnant women in Ethiopia. The results indicated that optimal access to ANC was high among pregnant women aged 25–34 (25.09%), those who had completed secondary and higher education (44.86%), and those who were working (26.24%). A greater proportion of women also get optimal ANC from a communities with high community media exposure (27.17%), lived in communities with high education (28.24%).

### Optimal access to ANC

Fig 1 displays the results of optimal access to ANC among pregnant women in Ethiopia. Of the pregnant women the majority (78.15%) had get skilled health care provider. The overall optimal access to ANC was 20.00% (18.93, 20.12).

**Table 2. Multilevel analysis of factors associated with optimal ANC access in Ethiopia, 2016 (n = 4771).**

| Variables | Optimal access to ANC | | Model 1 aOR [95% CI] | Model 2 aOR [95% CI] | Model 3 AOR [95% CI] |
|---|---|---|---|---|---|
| | Yes n(%) | No n(%) | | | |
| **Individual level factors** | | | | | |
| **Age in years** | | | | | |
| 15–24 | 281(22.73) | 956(77.27) | 1 | | 1 |
| 25–34 | 626(25.09) | 1870(74.91) | 1.75(1.36, 2.24) | | 1.58(1.23, 2.03)* |
| 35–49 | 235(22.64) | 804(77.36) | 2.42(1.71, 3.41) | | 2.04(1.43, 2.89)* |
| **Women educational status** | | | | | |
| No formal education | 509(19.74) | 2071(80.26) | 1 | | 1 |
| Primary | 510(26.60) | 1407(73.40) | 1.71(1.37, 2.12) | | 1.67(1.33, 2.09)* |
| Secondary and Higher | 123(44.86) | 152(55.14) | 2.25(1.45, 3.50) | | 1.81(1.15, 2.85)* |
| **Husband educational status** | | | | | |
| No formal education | 372(20.49) | 1445(79.51) | 1 | | 1 |
| Primary | 503(22.65) | 1737(77.35) | 1.05(0.85, 1.30) | | 1.04(0.84, 1.28) |
| Secondary and Higher | 163(38.86) | 256(61.14) | 1.45(1.01, 2.11) | | 1.31(0.88, 1.29) |
| **Occupation of women** | | | | | |
| Not working | 526(21.72) | 1895(78.28) | 1 | | 1 |
| Working | 617(26.24) | 1734(73.76) | 1.10(0.92, 1.32) | | 1.13(0.93, 1.31) |
| **Marital status** | | | | | |
| Unmarried | 117(34.31) | 225(65.69) | 1 | | 1 |
| Married | 1025(23.15) | 3404(76.85) | 0.61(0.29, 1.25) | | 0.76(0.36, 1.59) |
| **Religion** | | | | | |
| Orthodox | 626(30.84) | 1404(69.16) | 1 | | 1 |
| Muslim | 342(21.75) | 1231(78.25) | 0.68(0.53, 0.89) | | 0.88(0.66, 1.20) |
| Protestant | 153(14.61) | 896(85.39) | 0.39(0.28, 0.53) | | 0.65(0.44, 1.04) |
| Catholic and traditional | 21(17.62) | 98(82.38) | 0.47(0.21, 1.07) | | 0.71(0.31, 1.61) |
| **Wealth index** | | | | | |
| Poor | 328(18.94) | 1402(81.06) | 1 | | 1 |
| Middle | 213(21.40) | 783(78.60) | 1.10(0.85, 1.43) | | 1.09(0.83, 1.42) |
| Rich | 602(29.42) | 1444(70.58) | 1.44(1.12, 1.84) | | 1.16(0.88, 1.54) |
| **Media exposure** | | | | | |
| No | 543(19.80) | 2200(80.20) | 1 | | 1 |
| Yes | 599(29.56) | 1429(70.44) | 1.16(0.95, 1.42) | | 1.04(0.84, 1.29) |
| **Parity** | | | | | |
| 1 | 338(30.07) | 785(69.93) | 2.89(1.99, 4.20) | | 2.45(1.68, 3.59)* |
| 2–5 | 627(24.32) | 1949(75.68) | 1.69(1.27, 2.25) | | 1.48(1.11, 1.98)* |
| 5 and above | 178(16.63) | 895(83.37) | 1 | | 1 |
| **Decision maker for health service utilization** | | | | | |
| Women | 895(23.80) | 2867(76.20) | 0.99(0.78, 1.28) | | 0.95(0.74, 1.22) |
| Other | 148(20.63) | 571(79.37) | 1 | | 1 |
| **Community level factors** | | | | | |
| **Residence** | | | | | |
| Urban | 353(40.26) | 523(59.74) | | 2.62(1.74, 3.96) | 2.08(1.33, 3.27)* |
| Rural | 790(20.28) | 3106(79.72) | | 1 | 1 |
| **Distance to the health facility** | | | | | |
| Not big problem | 504(21.00) | 1897(79.00) | | 1 | 1 |
| Big problem | 638(26.93) | 1732(73.07) | | 0.76(0.62, 0.93) | 0.78(0.63, 0.96)* |
| **Community level media exposure** | | | | | |
| Low | 527(21.04) | 1980(78.96) | | 1 | 1 |

(*Continued*)

**Table 2.** (Continued)

| Variables | Optimal access to ANC | | Model 1 aOR [95% CI] | Model 2 aOR [95% CI] | Model 3 AOR [95% CI] |
|---|---|---|---|---|---|
| | Yes n(%) | No n(%) | | | |
| High | 615(27.17) | 1649(72.83) | | 1.22(0.89, 1.67) | 1.09(0.77, 1.54) |
| **Community level poverty** | | | | | |
| Low | 455(19.59) | 1868(80.41) | | 0.71(0.52, 0.92) | 0.79(0.56,1.12) |
| High | 688(28.08) | 1760(71.92) | | 1 | 1 |
| **Community level education** | | | | | |
| Low | 640(21.39) | 2351(78.61) | | 1 | 1 |
| High | 503(28.24) | 1278(71.76) | | 1.03(0.76, 1.41) | 0.82(0.58, 1.15) |
| **Region** | | | | | |
| Tigray | 146(30.14) | 339(69.86) | | 1 | 1 |
| Afar | 8(20.49) | 29(79.51) | | 0.54(0.20, 1.46) | 0.71(0.24, 2.09) |
| Amhara | 327(29.65) | 777(70.35) | | 0.86(0.58, 1.30) | 0.85(0.55, 1.32) |
| Oromia | 324(20.15) | 1283(79.85) | | 0.38(0.25, 0.57) | 0.48(0.29, 0.77)* |
| Somalia | 14(11.75) | 104(88.25) | | 0.28(0.13, 0.62) | 0.37(0.15, 0.88)* |
| Benshangul | 9(16.11) | 47(83.89) | | 0.34(0.13, 0.88) | 0.38(0.14, 1.05) |
| SNNPR | 174(15.64) | 940(84.36) | | 0.31(0.20, 0.47) | 0.39(0.23, 0.65)* |
| Gambela | 5(31.68) | 10(68.32) | | 0.74(0.20, 0.64) | 0.86(0.21, 3.47) |
| Harari | 5(33.9) | 9(66.10) | | 0.89(0.24, 3.32) | 1.10(0.27, 4.51) |
| Addis Ababa | 114(59.51) | 78(40.49) | | 1.67(0.97, 2.88) | 1.54(0.84, 2.80) |
| Dire Dawa | 17(57.73) | 12(42.27) | | 2.30(0.92, 5.78) | 3.07(1.09, 8.65)* |
| **Random effect** | | | | | |
| Variance | | 1.04 | 0.83 | 0.78 | 0.65 |
| ICC (%) | | 0.24 | 0.20 | 0.19 | 0.16 |
| MOR | | 2.63 | 2.36 | 2.29 | 2.13 |
| PCV | | ref | 20.14 | 25 | 37.5 |
| **Model comparission** | | | | | |
| Deviance(-2loglikelihood) | | 4948.28 | 4369.14 | 4782.76 | 4303.34 |
| Mean VIF | | *reference* | 1.33 | 1.44 | 1.59 |

Source: Demographic and Health Surveys

*P-value < 0.05, ICC: Intra class corrolation cofficent; MOR: Median odds ratio; PCV: Proportional change in variance; AOR: adjusted odds ratio; CI: confidence interval; VIF: Variance Inflation Factor

## Association between individual and community level factors and optimal access to ANC among pregnant women in Ethiopia

As shown in Table 2 both the individual and community level factors were significantly associated with optimal access to ANC among pregnant women in Ethiopia. With the individual level factors age, educational level, and parity were significantly associated with optimal access to ANC. The community level variables of residence, distance to the nearest health facility and region showed association with the phenomenon.

The odds of optimal access to ANC among pregnant women aged 25–34 years was 1.58 [aOR = 1.58, 95% CI = 1.23–2.03] and among 35–49 was 2.04 [aOR = 2.04, 95% CI = 1.43–2.89] times higher than those women aged between 15–24 years.

The odds of optimal access to ANC was 1.67 [aOR = 1.67, 95% CI = 1.33–2.09] and 1.81 [aOR = 1.81, 95% CI = 1.15–2.85] times higher among those women who had attended primary and secondary education than those women who had no formal education respectively.

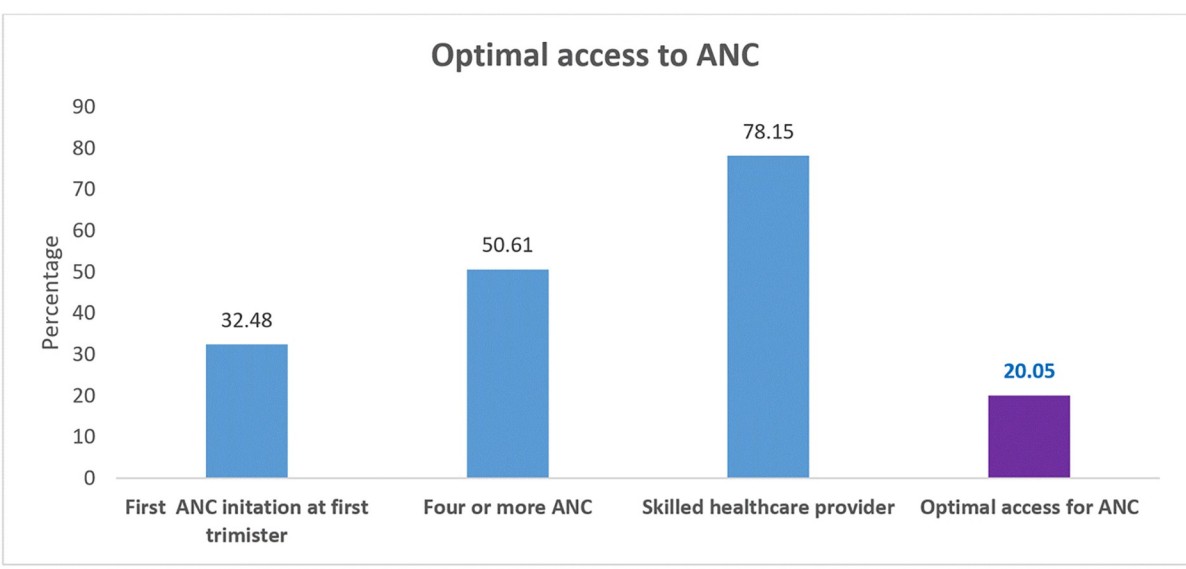

**Fig 1. Optimal access to ANC among pregnant women in Ethiopia, 2016 (n = 4771).**

The likelihood of optimal ANC access was 2.45 [aOR = 2.45, 95% CI = 1.68–3.59] and 1.48 [aOR = 1.48, 95% CI = 1.11–1.98] times higher among those women of parity 1 and 2–5 as compared to grand multipara women respectively.

Urban resident women had 2.08 [aOR = 2.08, 95% CI = 1.33–3.27] times higher odds of optimal access to ANC than rural resident women.

Those women who reported big problems regarding the distance to the nearby health facility had 0.78 times lower odds of [aOR = 0.78, 95% CI = 0.63–0.96] optimal access to ANC than those women who had reported the distance as not a big problem to the nearby health facility.

Those pregnant women from Oromia [aOR = 0.48, 95% CI = 0.29–0.77], Somalia [aOR = 0.37, 95% CI = 0.15–0.88] and South Nations Nationality regions [aOR = 0.39, 95% CI = 0.23–0.65] had 0.48, 0.37 and 0.39 times lower odds of optimal ANC access than women living in Tigray Regional state, respectively. On the other hand women from Dire Dawa administrative city had 3.07 [aOR = 3.07, 95% CI = 1.09–8.65] times higher odds of optimal access to ANC.

## Discussion

The study composed the variable optimal access to ANC, to understand multiple aspects of antenatal care using the timing of first ANC visit, adequacy of ANC visit and provider of care. Including these variables enable to measure both the availability and the use of services. The finding of this study revealed that one in five pregnant women had optimal access to ANC. On the other hand the majority (four in five pregnant women) did not either initiate their first ANC timely or had no four ANC visits, or not attended skilled ANC providers.

The finding is lower than the 2018 global report of 62%, SSA of 52% [39] and Cameroon (66%) [3]. The current study's lower prevalence may be related to the differences in the outcome variable construction. This study considered optimal ANC by combining the timing and number of ANCs as well as the skilled health care providers. Whereas the previous studies were considered optimal ANC through considering only number of ANC visit which may overestimate the prevalence. It is therefore necessary to introduce more innovative

approaches in order to increase the utilizations of recommended indicators of ANC in Ethiopia.

This study reveals that both the individual and community level factors were significantly associated with optimal access to ANC among pregnant women in Ethiopia. Accordingly, age of the women, educational status, and parity from the individual related factors and residence, region and distance to the health facility from the community related factors were significantly associated with optimal access to ANC.

The odds of optimal access to ANC among young women was higher than those older aged women. The finding is similar with studies conducted in Cameroon [3], and Sudan [40]. The possible justification for the higher optimal access, might be those older women are not expecting to become pregnant, leading to use less ANC visits as compared to young aged women [41]. Moreover, those older women might receive enough information about pregnancy related care from their lives, relatives, and media [42]. Therefore, improving optimal ANC access for older pregnant women might be very important to address maternal health for all reproductive age groups.

This study reveals that the odds of optimal access to ANC was higher among educated women than those women who had no formal education. The finding is similar with studies conducted in Cameroon [3], Nigeria [43] and Kenya [44]. The possible reason might be that more educated women are more likely to seek medical care from a trained professional. Women with a higher degree of education tend to have a better understanding of self-care, familiar with pregnancy-related issues. and have greater household decision-making power, and also know more about pregnancy-related complications and the benefits of recommended ANC services [45, 46]. This implies that empowering women through education leads to a better ANC follow up.

The likelihood of optimal ANC access was higher among those women of lower parity compared to grand multipara women. This finding is similar with studies conducted in Cameroon [3], Sudan [40]. This might be because of the grand multipara women are mostly feeling over confidence from their experience of previous pregnancy [47]. Additionally, women of multiparous are not expecting to become pregnant, leading to use less ANC visits [41].

This study reveals that urban resided women had higher odds of optimal access to ANC than rural resident women [3]. The finding is in congruent with a study conducted in Bangladesh [45] the poor optimal ANC in rural area might be that disparities are attributed to the transportation barrier and other supply-side constraints (such as the availability of facilities, health personnel, and results of diagnostic laboratory tests) [46]. Rural areas are often difficult to travel for pregnant women, especially if the roads are in poor condition. Moreover, developing countries, particularly rural areas, often have shortages of skilled attendants [48]. Therefore, Ethiopia may have achieved optimal coverage of ANC visits because of health insurance, free medical expenses, improved human resources, and road construction in rural areas [22].

The study finding reveals that distance to the health facility is an important factor for optimal ANC access. Those women who reported distance to the nearby health facility as big problems had lower odds of optimal access to ANC than those women who had big problems to the nearby health facility. The finding is similar with elsewhere studies in Uganda [49] and Tanzania [50]. The finding implies that access such as road construction and distribution of health services in remote areas needs the government attention.

The finding of this research revealed that there is differences for optimal ANC access across the country regions. Women living in Oromia, Somalia and South Nations Nationality regions had lower odds of optimal ANC access than women living in Tigray Regional state. The finding is supported by previous studies conducted in Ethiopia [23, 51]. Whereas those women from the Dire Dawa administrative city had higher likelihood to access optimal ANC. The

possible reason might be due to better accessibility of ANC in terms of transport, health workforce and medical supplies in urban areas. Dire Dawa is a city administration where a better accessibility for maternal care and women might have also better awareness about ANC.

### Strengths and limitations

As a strength, this study used an extensive national representative sample of data, advanced statistical models were applied to account for correlations within clusters. On the other hand, timing of fourth ANC might be very important but not considered in this study, if included it may underestimate the prevalence of optimal ANC. Women may mistakenly recall details after 5 years, and they may over report characteristics that are perceived as desirable. Leading to overestimation of proportions of women who fall into these categories. This measure does not consider gestational duration. The number of visits would therefore be considered insufficient for women who did not make it to the third trimester. Moreover, using four or more ANC visit instead of eight contact may lead to overestimation of the magnitude of optimal access.

### Conclusion and recommendation

This study concludes that only one among five pregnant women had optimal access to ANC. Both the individual level factors such as age, education, parity and community level factors of residence distance to the health facility and region were identified predictors for optimal ANC access. Therefore, to improve optimal ANC access in a comprehensive way, the Ethiopian government needs to improve women empowerment through extensive education. Moreover, attention for rural resident women and for those women who reported distance to the health facility as a big problem needs strong attention.

### Acknowledgments

For this study, we thank the MEASURE DHS programs for granting permission to use the relevant EDHS data.

### Declarations

**Ethics approval and informed consent**. During the original collection of DHS data, international and national ethical guidelines were taken into account. Ethical clearance for the original EDHS was approved by the Ethiopian Public Health Institute Review Board, Ethiopian Health and Nutrition Research Institute (EHNRI) Review Board, the National Research Ethics Review Committee (NRERC) at the Federal Democratic Republic of Ethiopia Ministry of Science and Technology, the ICF Macro Institutional Review Board, and the Centers for Disease Control and Prevention (CDC). According to the EDHS 2016 publications, all respondents provided written consent to participate. All the methods were conducted according to Helsinki declarations. No information obtained from the data set was disclosed to any third person. The study is not an experimental study. Further explanation of how the DHS uses data and its ethical standards can be found at: http://goo.gl/ny8T6X.

   **Consent for publication**. It is not applicable for this study since the study was used a secondary data analysis conducted by central statistical agency.

### Author Contributions

**Conceptualization:** Wubshet Debebe Negash.

**Data curation:** Wubshet Debebe Negash, Samrawit Mihret Fetene.

**Formal analysis:** Wubshet Debebe Negash.

**Software:** Tadele Biresaw Belachew.

**Supervision:** Samrawit Mihret Fetene, Banchilay Addis, Alebachew Ferede Zegeye, Tadesse Tarik Tamir.

**Validation:** Tadele Biresaw Belachew, Samrawit Mihret Fetene, Banchilay Addis.

**Visualization:** Tadele Biresaw Belachew, Banchilay Addis, Abel Endawkie, Sisay Maru Wubante, Elsa Awoke Fentie, Desale Bihonegn Asmamaw.

**Writing – original draft:** Wubshet Debebe Negash.

**Writing – review & editing:** Wubshet Debebe Negash, Tsegaw Amare, Atitegeb Abera Kidie, Abel Endawkie, Alebachew Ferede Zegeye, Tadesse Tarik Tamir, Sisay Maru Wubante, Elsa Awoke Fentie, Desale Bihonegn Asmamaw.

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
