## [Decision Letter · Decision Letter 0]

9 Feb 2023

PONE-D-22-32202One in five pregnant women had optimal access to ANC in Ethiopia: Multilevel mixed effect analysis of nationally representative cross sectional surveyPLOS ONE

Dear Dr.Negash,

Thank you for submitting your manuscript to PLOS ONE. After careful consideration, we feel that it has merit but does not fully meet PLOS ONE’s publication criteria as it currently stands. Therefore, we invite you to submit a revised version of the manuscript that addresses the points raised during the review process.  Please, see the reviewer comments below. 

We look forward to receiving your revised manuscript.

Kind regards,

Tesera Bitew, PhD

Academic Editor

PLOS ONE

Journal Requirements:

  "No funding was secured for this study."

Reviewers' comments:

Reviewer's Responses to Questions

**Comments to the Author**

1. Is the manuscript technically sound, and do the data support the conclusions?

Reviewer #1: Yes

Reviewer #2: Yes

2. Has the statistical analysis been performed appropriately and rigorously? 

Reviewer #1: Yes

Reviewer #2: Yes

3. Have the authors made all data underlying the findings in their manuscript fully available?

Reviewer #1: Yes

Reviewer #2: Yes

4. Is the manuscript presented in an intelligible fashion and written in standard English?

Reviewer #1: Yes

Reviewer #2: No

5. Review Comments to the Author

Reviewer #1: The study "One in five pregnant women had optimal access to ANC in Ethiopia: Multilevel mixed effect analysis of nationally representative cross sectional survey" showed that optimal access to antenatal care, such as first visit in the first trimester, four or more visits and a skilled health care provider, can significantly reduce maternal mortality in an inclusive manner. They also relate access to ANC to factors such as women's education, rural or urban location, age and country regions which offers a more comprehensive picture of the benefits of antenatal care. This is relevant to show the Ethiopian authorities the importance of improving the care centers, as well as the specialists in these centers, in order to reduce malnutrition, stillbirths, maternal mortality, and neonatal mortality. In spite of that there are previous studies with similar focus, and methodology such as "Tegegne, T. K., Chojenta, C., Getachew, T., Smith, R., & Loxton, D. (2019). Antenatal care use in Ethiopia: a spatial and multilevel analysis. BMC pregnancy and childbirth, 19(1), 1-16". In my opinion, the fact that the authors included the variables such as: time of first ANC visit and type of ANC provider it may offer a better picture of the benefits of Antenatal care and offer characteristics that the government of Ethiopia should prioritise in the care of pregnant mothers and new-borns.

The manuscript is well prepared, but there are some minor corrections that need to be carried out, which are outlined below. The introduction is appropriate and focuses on the scope of the study. The methodology is well described, indicating how the dependent variable "optimal ANC access" was obtained, and the different levels of the dependent variables. In addition, multilevel logistic regression is adequate to analyse the association between the dependent variables and the independent variables. Likewise, multilevel mixed-effects logistic regression is suitable for evaluating the association between the optimal access to ANC and the independent variables. However, it is important to detail which variables were considered as fixed and mixed effects. In addition, i would only suggest that the odd-ratio be presented in a forest plot and the intra-class correlation coefficient in a scatterplot. The discussion is also adequate, addressing each of the variables analysed and their odd ratio to optimal access to ANC.

Minor considerations

Line 166. - Please indicate on this line that the result of this logistic regression is presented in Table 1.

Line 169. - 170. - The authors mention that the graph of optimal ANC access among women of reproductive age was presented, but this graph is not indicated in the manuscript, nor is it indicated whether the graph is in the supplementary material.

Line 177 - 178.- Indicate which variables were considered as fixed effect and random effect in the final model.

Line 194 - 197.- I suggest for a better understanding that in this section you include the variables that were considered in the final model and create a table showing only the models and their goodness-of-fit values for each of the three proposed models.

Line 207. - I do not understand why the authors place table 1 inside the manuscript and table 2 in the tables (Line 503) and figures section. Please correct this.

Line 222.- I believe you are referring to the range of pregnant women 35-49 years of age.

Line 223.- I believe you are referring to the range of pregnant women 15-24 years of age.

Line 594.- I suggest that only the results of the final model be presented in Table 2 for a clearer presentation of the results. Since it is assumed that the other models are not considered and the data presented are not discussed. Another option would still be to keep this entire table 2, but in the supplementary material section.

Reviewer #2: Review response for the title ‘’One in five pregnant women had optimal access to ANC in Ethiopia: Multilevel mixed effect analysis of nationally representative cross-sectional survey’’

By Biresaw Ayen Tegege

Dear inviter Dr. Tesera Bitew, Thank you for giving me the opportunity to review this manuscript.

Dear authors

Thank you very much for your effort and relevant analysis in the field of Optimal ANC access in Ethiopia.

Please find my suggestions for major revisions below.

Step-by-step suggestions:

Major points

1.Title: your title seems the result of study, it would be good if you can say the magnitude of optimal access to ANC and its predictors in Ethiopia: Multilevel mixed effect analysis of nationally representative cross-sectional survey.

2016 WHO ANC model recommended minimum of eight ANC visits but you want to investigate on under dated and limited outcome (4 and above). Why?

So that it is difficult to conclude as 20 %had optimal access to ANC.

Furthermore, your title is not inline with your objective, please make it inline

2. Keywords: you have used key words like Factors…..but It is better to avoid such terms.

3. Result

Your table 1 has showed that the association is only descriptive and vague

Could you able to present all Models with its respective adjusted odds ratio in tabular form?????

I could not find table 2 in the result section

You wrote table 1 at result section and table 2 at the end why?

Can you please reform tables as individual and community level variables?

5. Discussion

Writing number, % …. not important here since it has been written at result section

6. Limitation

It would be good if you write the use of four and above ANC access as cut point in your study as limitation.

7. Recommendation

Your recommendation is inside the conclusion, can you write the heading as conclusion and recommendation?

6. PLOS authors have the option to publish the peer review history of their article (what does this mean?). If published, this will include your full peer review and any attached files.

Reviewer #1: No

Reviewer #2: No

---

## [Author Response · Author response to Decision Letter 0]

14 Feb 2023

Authors’ Point-by-point response to editor and reviewer comments 

We are very grateful to both the editor and reviewers for your comments and suggestions. All the concerns raised so far will have an undeniable impact on improving the quality and readability of our scholarly work. Appreciating your effort and valuable comments, we have provided possible reflections and amended the raised concerns and questions. Kindly find our reflections here.

We hope you will consider the revised manuscript acceptable for publication in PLOS ONE research.

S.no Reviewers’ comments Authors’ responses 

 Review Comments to the Author 

 Reviewer #1 

1

 The study "One in five pregnant women had optimal access to ANC in Ethiopia: Multilevel mixed effect analysis of nationally representative cross sectional survey" showed that optimal access to antenatal care, such as first visit in the first trimester, four or more visits and a skilled health care provider, can significantly reduce maternal mortality in an inclusive manner. They also relate access to ANC to factors such as women's education, rural or urban location, age and country regions which offers a more comprehensive picture of the benefits of antenatal care. This is relevant to show the Ethiopian authorities the importance of improving the care centers, as well as the specialists in these centers, in order to reduce malnutrition, stillbirths, maternal mortality, and neonatal mortality. In spite of that there are previous studies with similar focus, and methodology such as "Tegegne, T. K., Chojenta, C., Getachew, T., Smith, R., & Loxton, D. (2019). Antenatal care use in Ethiopia: a spatial and multilevel analysis. BMC pregnancy and childbirth, 19(1), 1-16". In my opinion, the fact that the authors included the variables such as: time of first ANC visit and type of ANC provider it may offer a better picture of the benefits of Antenatal care and offer characteristics that the government of Ethiopia should prioritise in the care of pregnant mothers and new-borns. Dear reviewer, thank you for your insights and appreciation. We have accepted your comments.

2 The manuscript is well prepared, but there are some minor corrections that need to be carried out, which are outlined below. The introduction is appropriate and focuses on the scope of the study. The methodology is well described, indicating how the dependent variable "optimal ANC access" was obtained, and the different levels of the dependent variables. In addition, multilevel logistic regression is adequate to analyse the association between the dependent variables and the independent variables. Likewise, multilevel mixed-effects logistic regression is suitable for evaluating the association between the optimal access to ANC and the independent variables. However, it is important to detail which variables were considered as fixed and mixed effects. In addition, I would only suggest that the odd-ratio be presented in a forest plot and the intra-class correlation coefficient in a scatterplot. The discussion is also adequate, addressing each of the variables analysed and their odd ratio to optimal access to ANC. Dear reviewer thank you for your evaluation. The variables considered as fixed and random effects are revised at page10 line 178- 180. 

For your suggestion of “I would only suggest that the odd-ratio be presented in a forest plot and the intra-class correlation coefficient in a scatterplot.” We complain that the current presentation is better than the forest plot. Because the forest plot is used for estimating the pooled results and heterogeneity of all results from a number of scientific studies (or in short used for summarizing results from different individual studies). Forest plots should not be generated that contain no studies, and are discouraged when only a single study is found for a particular outcome1,2. Generally, forest plots are recommended when

1. There are sufficient studies to make them of value 

2. Ensure that plots contain all of the important elements

3. Exploit the plot's vertical dimension by ordering studies in a way that might illustrate important differences among them, such as by year of publication, effect size, or important study characteristic.

On the contrary the current study is conducted in a single country Ethiopia. Therefore, for a single country it could be better to explain the value of odds ratio and intra-class correlation as it is.

 Minor considerations 

3 Line 166. - Please indicate on this line that the result of this logistic regression is presented in Table 1. Thank you, now indicated. Kindly see the revised manuscript. (Page 9; lines 168)

4 Line 169. - 170. - The authors mention that the graph of optimal ANC access among women of reproductive age was presented, but this graph is not indicated in the manuscript, nor is it indicated whether the graph is in the supplementary material. Indicated in the manuscript. Kindly see the revised manuscript. (Page 14; lines 215)

5 Line 177 - 178 Indicate which variables were considered as fixed effect and random effect in the final model. Thank you for your gentle observation. All the individual and community level variables were considered as fixed effect, whereas, cluster number was the random effect variable. Based on the random variable the measure of variation such as ICC, MOR, and PCV were calculated. As per your comments the part is revised in the clean version of the manuscript (page10 line 178- 180.)

6 Line 194 - 197.- I suggest for a better understanding that in this section you include the variables that were considered in the final model and create a table showing only the models and their goodness-of-fit values for each of the three proposed models. Dear reviewer, in fact your suggestion is very good. However, this is the methods section, so that the table showing the models goodness of fit for each of the three models is presented in the result section at the end part of table 2.

Kindly see table 2 

7 Line 207. - I do not understand why the authors place table 1 inside the manuscript and table 2 in the tables (Line 503) and figures section. Please correct this. Thank you for your question. The reason why the authors were place table two and figure in the tables and figures section was to follow the journal guideline. Now it is corrected as per your comments. Kindly find table 1, 2 and figure in the manuscript.

8 Line 222.- I believe you are referring to the range of pregnant women 35-49 years of age. Yes! Kindly see the clean version of the manuscript 

Page 15, line 225 

9 Line 223.- I believe you are referring to the range of pregnant women 15-24 years of age. Yes! Kindly see the clean version of the manuscript 

Page 15, line 227

10 Line 594.- I suggest that only the results of the final model be presented in Table 2 for a clearer presentation of the results. Since it is assumed that the other models are not considered and the data presented are not discussed. Another option would still be to keep this entire table 2, but in the supplementary material section. Dear reviewer your suggestion might be very significant but other reviewers are suggested to present all Models with its respective adjusted odds ratio in tabular form with in the manuscript. Even though other models are not discussed, their presentation is important for visual description and comparison.

 Reviewer #2 Authors response

1 Title: your title seems the result of study, it would be good if you can say the magnitude of optimal access to ANC and its predictors in Ethiopia: Multilevel mixed effect analysis of nationally representative cross-sectional survey. Dear reviewer thank you for your interesting observation. The title is changed as per your comments. Kindly see the clean version of the manuscript.

Page 1 line 1-2

2 2016 WHO ANC model recommended minimum of eight ANC visits but you want to investigate on under dated and limited outcome (4 and above). Why? So that it is difficult to conclude as 20 %had optimal access to ANC. Dear reviewer, in fact you are correct that the 2016 WHO recommends a minimum of eight contacts. However, many African countries including Ethiopia are trying to achieve high coverage of focused ANC. Ethiopia is still using the previous four ANC to declare a woman as having optimal ANC visit 3-6. In these four contacts, packages such as identifications of preexisting health condition, early detection of complications related with pregnancy, health promotion and complication readiness are carried out. We included this cut off point in the limitation section. Kindly see the clean version of the manuscript on Page 23 line 323-324

3 Furthermore, your title is not in line with your objective, please make it inline Thank you. Now the title is made in line with our objective. Kindly see clean version of the manuscript Page 1 line 1-2

4 Keywords: you have used key words like Factors…..but It is better to avoid such terms. Avoided. Kindly see the clean version of the manuscript 

Page 4 line 66

 Result 

5 Your table 1 has showed that the association is only descriptive and vague

Could you able to present all Models with its respective adjusted odds ratio in tabular form????? Dear reviewer thank you for your comments. All models with its respective adjusted odds ratio is presented in tabular form in table 2. Kindly see table 2.

6 I could not find table 2 in the result section

You wrote table 1 at result section and table 2 at the end why?

 Thank you. Now we corrected the position of the tables accordingly. Kindly see table 1 and 2 in the clean version of the manuscript.

7 Can you please reform tables as individual and community level variables? The tables are reformed. Kindly see table 1 and 2

 Discussion 

8 Writing number, % …. not important here since it has been written at result section

 Really thank you for your comments. We have revised the discussion section. Kindly see the clean version of the manuscript page 21 line 252-258

9 Limitation

It would be good if you write the use of four and above ANC access as cut point in your study as limitation. Thank you for your insight comment. The limitation is included kindly see the clean version of the manuscript on page 23 line 323-324

10 Recommendation

Your recommendation is inside the conclusion, can you write the heading as conclusion and recommendation? Corrected. Kindly see the clean version of the manuscript on page 23 line 325

References 

1. Schriger DL, Altman DG, Vetter JA, Heafner T, Moher D. Forest plots in reports of systematic reviews: a cross-sectional study reviewing current practice. International journal of epidemiology. 2010;39(2):421-429.

2. Lewis S, Clarke M. Forest plots: trying to see the wood and the trees. Bmj. 2001;322(7300):1479-1480.

3. Organization WH. WHO antenatal care randomized trial: manual for the implementation of the new model: World Health Organization;2002. 9241546298.

4. Organization WH. WHO recommendations on antenatal care for a positive pregnancy experience: summary: highlights and key messages from the World Health Organization's 2016 global recommendations for routine antenatal care: World Health Organization;2018.

5. Tessema ZT, Animut Y. Spatial distribution and determinants of an optimal ANC visit among pregnant women in Ethiopia: further analysis of 2016 Ethiopia demographic health survey. BMC pregnancy and childbirth. 2020;20:1-13.

6. Tekelab T, Chojenta C, Smith R, Loxton D. Factors affecting utilization of antenatal care in Ethiopia: a systematic review and meta-analysis. PloS one. 2019;14(4):e0214848.

---

## [Decision Letter · Decision Letter 1]

11 Apr 2023

Magnitude of optimal access to ANC and its predictors in Ethiopia: Multilevel mixed effect analysis of nationally representative cross-sectional survey

PONE-D-22-32202R1

Dear Dr. Negash,

We’re pleased to inform you that your manuscript has been judged scientifically suitable for publication and will be formally accepted for publication once it meets all outstanding technical requirements.

Kind regards,

Mulu Tiruneh

Academic Editor

PLOS ONE

Additional Editor Comments (optional):

Reviewers' comments:

Reviewer's Responses to Questions

**Comments to the Author**

1. If the authors have adequately addressed your comments raised in a previous round of review and you feel that this manuscript is now acceptable for publication, you may indicate that here to bypass the “Comments to the Author” section, enter your conflict of interest statement in the “Confidential to Editor” section, and submit your "Accept" recommendation.

Reviewer #1: All comments have been addressed

Reviewer #2: All comments have been addressed

2. Is the manuscript technically sound, and do the data support the conclusions?

Reviewer #1: Yes

Reviewer #2: Yes

3. Has the statistical analysis been performed appropriately and rigorously? 

Reviewer #1: Yes

Reviewer #2: Yes

4. Have the authors made all data underlying the findings in their manuscript fully available?

Reviewer #1: Yes

Reviewer #2: Yes

5. Is the manuscript presented in an intelligible fashion and written in standard English?

Reviewer #1: Yes

Reviewer #2: Yes

6. Review Comments to the Author

Reviewer #1: The authors have resolved the observations arising from the first revision. The manuscript has improved considerably and I consider it appropriate for journal publication. I have no other considerations.

Reviewer #2: I want to appreciate the efforts made by the authors to address the raised questions! well addressed and thank you!

7. PLOS authors have the option to publish the peer review history of their article (what does this mean?). If published, this will include your full peer review and any attached files.

Reviewer #1: No

Reviewer #2: No

---

## [Editor Report · Acceptance letter]

12 Apr 2023

PONE-D-22-32202R1 

Magnitude of optimal access to ANC and its predictors in Ethiopia: Multilevel mixed effect analysis of nationally representative cross-sectional survey  

Dear Dr. Negash:

I'm pleased to inform you that your manuscript has been deemed suitable for publication in PLOS ONE. Congratulations! Your manuscript is now with our production department. 

Kind regards, 

on behalf of

Mr Mulu Tiruneh 

Academic Editor

PLOS ONE